



# Application of advanced composite modified perlite for degradation of particle size and turbidity in treatment of sewage water

**Ali Reza Taheri Fard**[*1]

*Peter the Great St. Petersburg Polytechnic University,29 Politechnicheskaya St., St. Petersburg, 195251,Russia*

**Abstract:**

Water treatment efficiency of several filter media such as perlite, modified (silicated) perlite, zeolite, and sand were studied on sewage water. It was shown that modified perlite removed more than 90% of turbidity and it functioned more efficient than other materials in case of high turbidity (more than100 NTU). Filtration through modified perlite significantly decreased the concentration of total nitrogen (from 4 to 1 mg/L), chemical oxygen demand (from 274 to 0.42 MgO/L), concentration and size of particles (from 3870 nm to 56 nm). Filtering device was created with 2 steps syphon, due to having sedimentation part in the bottom part of next part and having two times upward direction in filtration undoubtedly operates better than monolayer filter with mere perlite. The total cost of filtration unit containing whole part of filtration device and advanced composite modified perlite materials as well as evaluates reducing the cost up 12% compared to ceramic filter.

**Keywords:** Water Treatment; Filtration; Modified perlite; particle size; total nitrate; NTU

## 1. Introduction:

The used process for water treatment is reliant upon quality of water sources. The surface water merely possesses more variant rather than groundwater in case of pollutants. Hence, the process might be complex for this type of water. Majority of surface water has turbidity more than the standards for the potable water. Although high-speed moving water may have larger suspended materials, but most of the solids are colloid, therefore, chemical separation and filtration (sedimentation) are required to separate them (Qi, 2013). To meet the needs of people and to provide access to clean water and, in the twentieth century is difficult and complicated. Global growth in the public water supply, improve water quality, climate change and are growing rapidly. The need of modern technology to ensure the integrated management of water resources (Thakare, Y.., 2013; Irani, M.,2011; Kucharczyk, W., 2017)Filtration or water treatment means remove solids from a liquid by passing it from a lattice or reticular chamber that these nets have very small openings (Li, L., 2017). Filtration is a main part of many industries such as the major chemical industries (sulfuric acid production or caustic soda producers), water purification, and food and beverage industry (Dempsey, M., 2005). At the present stage, the problem of deep purification of waters polluted remains relevant not only in connection with the small values of their MPCs (0.01.0.05 mg / l for fish-breeding reservoirs and 0.3 mg / l for cultural

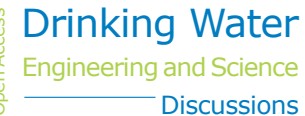

and household purposes and for drinking purposes), but also because they are represented in the drains
of almost all industrial enterprises (Bastani, D., 2006)
In the various branches of the filtration industry, the purpose is to obtain liquids with the highest
purity in a shorter time and lower cost, which is not possible without proper filtering (Majouli, A.,
2011).  Perlite is a type of rock that is a volcanic rock of volcanic glass which if heated sufficiently,
expands from 4 to 30 times its initial size (Guo, J. 2015). From the middle of the third century BC,
humans recognized this matter as a volcanic glass. The perlite originates from the word perl means
pearl (French word) (Vatin N.I., 2014). The perlite expands under the heat is due to 3-6% water in the
structure of perlite rocks. By heating perlite, the water in the structure and porosity of perlite rocks is
evaporated and millions of very small bubbles are formed in the perlite structure (Kim A., 2015).
Thus, the structure of perlite is transformed into a porous structure with closed cells, the volume of
its grains is strongly increased and its color changes from black or dark gray to white. Perlite consists
of silica oxide, aluminum, sodium, calcium, potassium, manganese, which is silica oxide possesses the
highest percentage among other ingredients. The expanded perlite is chemically neutral and after
production it becomes completely dry as well as white color (Andrianova M., 2014). The structure of
perlite is interconnected with tens of thousands of microscopic channels. This material provides an
optimal flow rate. The same properties of perlite make it very effective in purifying food, drinking and
medicine (YANG, G., 2007).
According to the report, application of perlite backs to 1800, but in the 1940s (Rodriguez, J., 2016;
Hagner, A., 1950; Tsikouras, B., 2016) the United States did not use the conditions of modern day, it
should have been used more than 2300 years ago (Annadurai, G., 2014). Perlite is found in many
countries around the world in 2011(Brown TJ, 2013), 95% of the total around 3.5 million tons of
global perlite production takes place in 10 countries, which the largest perlite makers are in in Iran,
China, Iran, Greece, Japan and Turkey.
Advantages of perlite as a filter aid include reducing the cost with the help of perlite filter which
implies the aid of perlite filter is 20 to 50% more than other filters, which is very significant in terms
of cost reduction. The perlite filter density is only 110 to 270kg / m$^3$ (Wyatt, A. 2004) Experience
shows that in using this filtering aid in place of other filtering assistance in different industries leads to
plunge the cost of refining without reducing efficiencies in the refinement (CHANG, S., 1989).
High transfer rate which is due to the perlite's unique physical structure, perlite filter contours help
high-fluid transfer rates of high quality. It is particularly helpful for highly viscous liquids such as
syrups or gelatinous fluids that require fast flow (Semra Siber Uluatam, 2007). Simple cleaning of the
mold which is the aid of the perlite filter, since this type of filter is porous and not compressed, it can
be easily cleaned after the work is completed, which will help reduce manpower and increase
productivity (Gironás, J., 2008). Non-hazardous waste which perlite is not a dangerous waste and can
be thrown away easily. Some molds that are used in the food process can be recycled even as part of
livestock feed (Uluatam, S.1992; Adams, F., 2017).



Application of perlite in filtration process has not been used for over a decade, since there has not
been proposed brand new idea and viewpoint on the basis of perlite in filtration, the proposed material
so called advanced composite modified perlite has undergone a hundred experiments to assert a
credence of it, suggested materials are totally new idea on the basis of filtration process, which would
make significant breakthrough in scientific area of filtration. While researchers are struggling with the
membranes, the advanced composite modified perlite will light up this domain of work with its span-
new performance.

**2. Materials:**

Input data and materials for the experiments contain polluted water such as sewage water or
artificially polluted water with clays or other materials. Moreover, using perlite was the main purpose
of this research which includes modified perlite with sodium silicate in certain process of producing,
modified perlite with synthetic zeolite and normal perlite with Fuler's curve grading. The result and
expected output of the work is to remove 90 percent of turbidity from water to obtain the range of
world water standard for drinking.

*2.1 Perlite:*

Perlite, especially expanded perlite has unique adsorbing properties. This is due to a significant
absorption surface and high adhesion properties of the material. Suffice it to say that this sorbent is
able to absorb the amount of material that surpasses it in volume from 4 to 20 times. At the same time,
a high porosity index (from 70 to 80%) causes a record absorption rate, and a small relatively small
pore diameter allows to retain the smallest particles of suspensions and liquids that need to be
collected.
These properties that have led to the widespread use of expanded perlite as a sorption material in
the collection of various liquids and as a filter.
The properties of perlite are widely used in the creation on its basis of portable and stationary
filters for wastewater treatment from mechanical impurities and oils, as well as pre-treatment of water
before usage.
The used perlite was taken from Zanjan Perlite Company with following characteristic in table 1 and
the physical characteristics of taken perlite is shown in table 2.
Table 1:characteristic of using perlite

| substance | Perlite % |
|---|---|
| SiO2 | 72.10 |
| Al2O3 | 12.95 |
| Fe2O3 | 0.88 |
| K2O | 3.92 |
| Na2O3 | 3.16 |
| H2O | 3.88 |

Table2: Physical characteristic of taken perlite

| color | white |
|---|---|





| pH | 6.5-7.5 |
|---|---|
| Specific gravity (gr/cm3) | 2.2-2.4 |
| Bulk density (Kg/m3) | 40-200 |
| Toughness (Mohs) | 5-6 |
| Specific heat capacity (J/Kg.K) | 837 |
| Heat transfer (W/m.K) (in 24 °c) | 0.04-0.06 |


*2.2 SC Perlite:*
Sodium silicate is originated from salts of silicon acids. For the first time liquid glass in 1818 was
received by the German chemist Jan Nepomuk von Fuchs. This compound is very widespread in
nature. Silicates are contained in one third of all known mineral compounds (in clay materials,
feldspar, and mica).
Sodium silicate is a white or whitish fine powder with no specific taste and odor. When Liquid
glass dissolves in water, forms a viscous solution. In dilute solutions, sodium silicate decomposes into
anions of silicic acid and sodium cations. When water is removed, the sodium silicate solution
becomes an amorphous solid. Under the action of chlorides and acids, a silica gel (sorbent) is formed
from the solution of the liquid glass. Viscous solutions of sodium silicate when heated to a
temperature of 200-300 ° C are swollen and increase in volume by a factor of five to eight.
At present, liquid glass is obtained by the method of autoclaving raw materials containing silica,
concentrated solutions of sodium hydroxide. Methods are also known for the production of sodium
silicate, based on the crystallization of melts from glasses, precipitation from the gas phase and
solutions.The sodium silicate went through mixing with perlite with Ratio 1:0.75, after 3 minutes
constant mixing sodium silicate with perlite, they were placed into oven in 100 degree Celsius for 24
hours in order to dry sodium silicate and form a new material for filtration treatment. The physical
properties of sodium silicate is represented in table 3:
Table 3: shows physical properties of sodium silicate

| Weight Na2O % | 8.9 |
|---|---|
| Weight SiO2 % | 28.7 |
| Weight solids % | 37.6 |
| Density (g/cm3) | 1.38 |


*2.3 Synthetic Zeolite covered perlite:*
Zeolites - a large group of similar in composition and properties of minerals, water aluminosilicates
of calcium and sodium from a subclass of framework silicates, with a glass or pearlescent shine. Their
main difference is that they absorb and emit not only water, but also other different molecules without
changing the crystal structure. Absorption of molecules by zeolites is associated with the phenomenon
of adsorption - the concentration of a substance from the gas phase on the surface of a solid
(adsorbent) or in the volume formed by its pore structure.
The use of natural zeolites was limited due to their low adsorption capacity, they were used for gases
and liquids with small molecule impurities, so they were used only to reduce water hardness. The
situation changed when, in the 1950s, the first synthetic zeolites were obtained in R. Barrera's





laboratory. Studies have shown that artificially synthesized zeolites as adsorbers have unique
properties, since they are capable of absorbing all components of complex mixtures. Also, they are
able to purify substances even from a small amount of undesirable impurities, which is very important
for some types of industry.


*2.4 Perite with modified Fuller'curve:*
Perhaps it is reasonable to believe that the best rating is which creates the maximum density. This
involves the accumulation of particles in which smaller particles are packed between larger particles
that reduce the free space between particles. This result in more particulate particles that increase
HMA stability and reduce water penetration. In PCC, provided that, the free space decreases, reducing
the amount of cement paste required. However, the minimum number of holes is required to ensure
rapid drainage and cold resistance for base and sub-base courses.
I used modified Fuller Thompson Curve with ratio of 0.6 to have bulk and dense perlite size in
filtration device, in table no 4 and Fig. 1 is shown the properties of perlite after using Fuller curve:
Table 4: shows properties of perlite after using Fuller curve

| No | size | remained | passing | Remained on each sieve (%) | weight on each sieve (gr) | density |
|---|---|---|---|---|---|---|
| #4 | 4.75 | 0 | 100 | 0 | 0.00 | 176 |
| #8 | 2.38 | 34 | 66 | 34 | 68.44 | 149 |
| #12 | 1.68 | 47 | 53 | 12 | 49.86 | 295.5 |
| #20 | 0.85 | 65 | 35 | 18 | 22.20 | 91 |
| #50 | 0.3 | 81 | 19 | 17 | 39.91 | 177 |
| Pan | 0.01 | 100 | 0 | 19 | 80.81 | 320 |







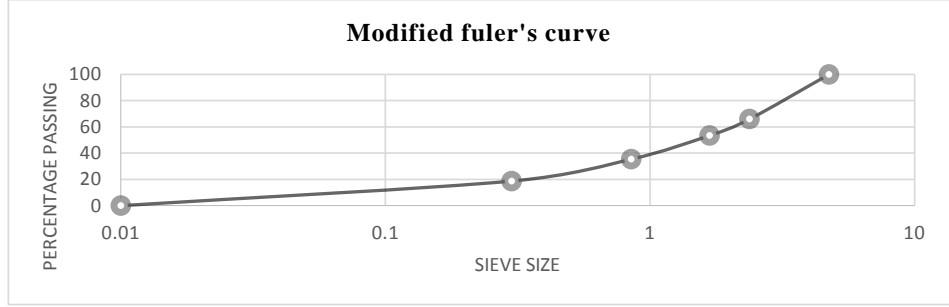





160                         Fig. 1: shows the graph of modified fuller curve in power of 0.6




*2.5 Activated carbon covered perlite:*
Activated charcoal operates through a process called adsorption, in which contamination molecules
are contained in a fluid treated in a carbon-based airway structure. A carbon filter is generally used for
water purification, air purification and industrial gas treatment, for example, removal of siloxanes and
hydrogen sulphide from biogas. It is also used in many other applications, such as breathing masks,
sugar cane cleaning and mining of precious metals, especially gold. It is also used on signal filters.
*2.6 Non-woven fabric:*
The particle size is the critical diameter of the solid spherical particles passing through these pores.
This minimum cavity is not the same for all color materials. Consequently, the change in particle size
is also contemplated. The greater number of layers of fibers and the like, nonwoven, are more likely to
be hit at least once with fabrics of minimal purity. Unparalleled homogeneity, the smallest difference
is the difference between the smallest diameter and the larger air outlet. Filters for metal needles / dyes
are usually thick and the adhesive fabrics are quite thin, but they have excellent filtration. There are
five different types of non-metallic processes used to make filter materials.
*2.7 proposed filtration device:*
It is good to have a filtration device which works upward and downward at the same time. In
addition, this device is working with Siphon law, and it is consist of 4 cylinders which 2 by 2 are
inside each other.
There are two leading theories about how siphons cause liquid to flow uphill, against gravity,
without being pumped, and powered only by gravity. The traditional theory for centuries was that
gravity pulling the liquid down on the exit side of the siphon, resulted in reduced pressure at the top of
the siphon. Then atmospheric pressure was able to push the liquid from the upper reservoir, up into the
reduced pressure at the top of the siphon, like in a barometer or drinking straw, and then over.
*2.8. Ceramic filtration:*
Filtration device was from company name AquaSafe with following description: The Doulton
Ultracarb is a three stage cartridge combining the highly efficient filtration properties of ceramic with
the enhanced water treatment properties of activated carbon and the heavy metal reduction capabilities
of ion exchange media.

**3. Experiments and results:**
*3.1 Turbid meter (NTU):*



The turbidity is relatively close to the amount of light scattered by 90 degrees when the light
source appears through the sample. During the measurement, the relationship between optical
dispersion and opacity is used to measure carbonate measurements of liquid samples. The experiments
were carried out with two samples of waste water. The result is shown in table5.
*3.2 TOC:*
Total carbon pollution (TOC) is the amount of carbon in an organic compound and is often used as
an indirect indicator of water quality or purity of pharmaceutical production equipment. TOC can also
refer to the amount of soil pollutants in the soil.
When carbon dioxide is oxidized and / or when inorganic carbon is acidic, then almost everyone
analyzes the TOC $CO_2$. Oxidation is done by catalytic combustion of PT, hot microbial or ultraviolet /
condensate reactor. When $CO_2$ is formed, then it is measured with a detector: a conductive element (in
the case of $CO_2$ $CO_2$) or a non-expansion infrared cell (after removal of $CO_2$ with water in the gas
phase). It is desirable to determine the conductivity in the range of low TOC in deionized water, while
TOC is best for detecting NDIR. The variety described in the form of membrane canmetry can be used
to measure the TOC in the analysis of the wide range of decay and non-deionized water samples. Very
effective synchronous TOCs are able to detect carbon concentrations below 1 μg / l (1 ppm or
peptide). The results is shown in table no5.
*3.3 Nitrate:*
Nitrate test is a chemical test that determines the presence of nitrate ions in a solution. Testing for
the presence of nitrate in wet chemistry is difficult compared to the comparison of other ions, because
almost all nitrate are dissolved in water. On the contrary, many common ions, insoluble salts, for
example, tend to strain with barium with the helms, silver and sulfate.
Table 5: shows the amount of TOC and COD after filtration and nitrate and NTU.

| Vial | sample | TOC | TN | TC | IC | COD | NTU | pH |
|------|--------|-----|-----|-----|-----|-----|-----|-----|
| 1 | Sewage water | 102.6 | 3.607 | 139.7 | 37.12 | 273.6 | 1052.652 | 5.7 |
| 2 | Modified perlite | 0.16 | 0.8723 | 20.51 | 20.35 | 0.42 | 4.556545 | 8.3 |
| 3 | Ceramic Filtration | 26.4 | 0.9317 | 70.14 | 43.61 | 70.4 | 60.41508 | 7.1 |


*3.4 Optical Density:*
In the total logarithm of chemistry, land improvement is parallel to the logarithm, event of radial
forces, particle object, and spectral gradient or spectral gradient as well as radial power spectrum.
There is no size of calculation and rendering any difference. The monotonic mode appears at different
levels and estimates the number by zero.
The optical density test has been conducted three times the first one was normal water, the second
experiment was advanced composite modified perlite with 10 times dilution with water  and the third
one was carried out through ceramic filtration with 10 times dilution with water. The results from
optical density of samples are demonstrated in Figs, 2, 3 and 4:



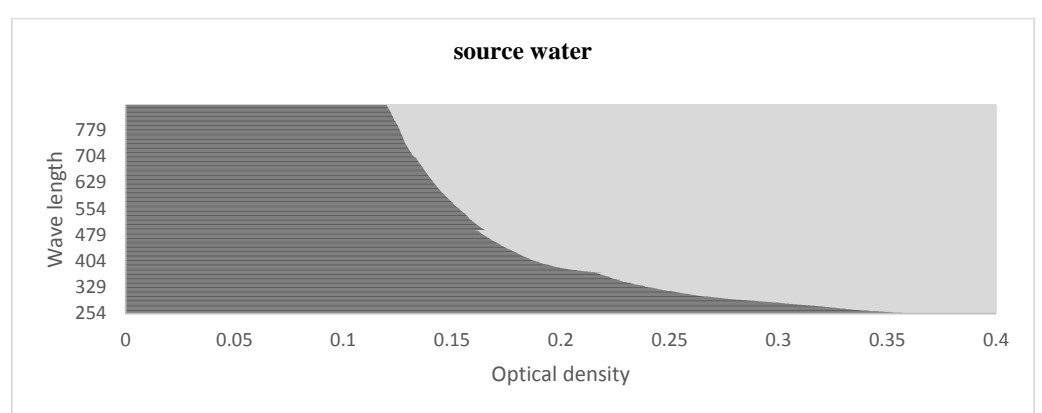


Fig. 2: the experiment of OD with source water diluted 10 times.




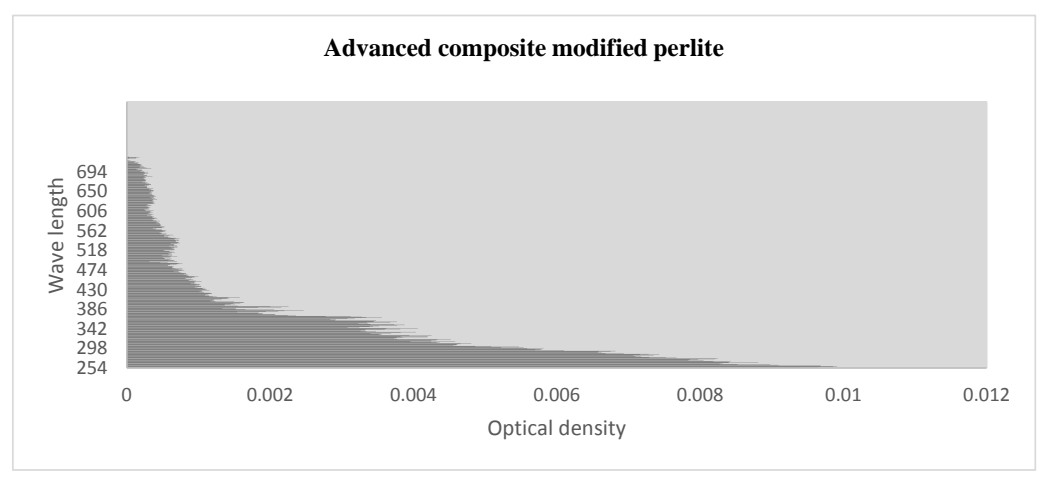






Fig 3: the experiment of OD with modified perlite filtration diluted 10 times



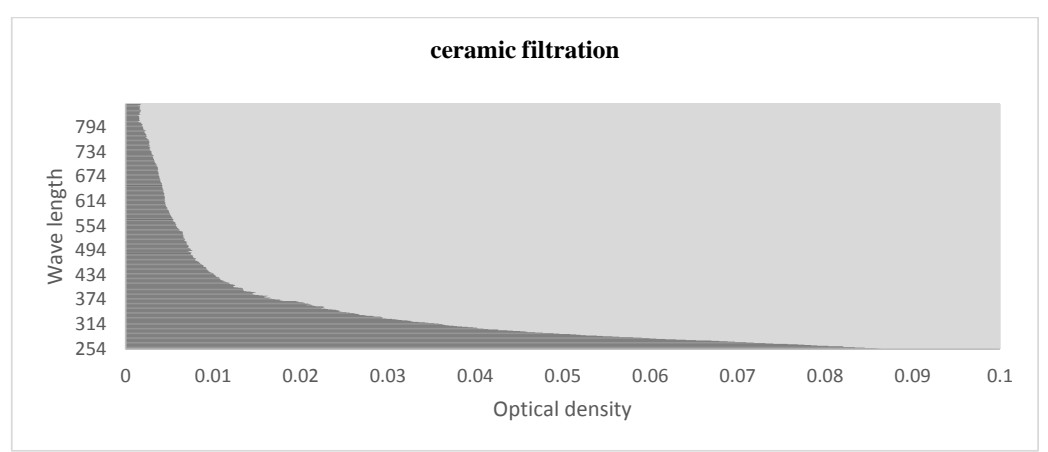


Fig 4: the experiment of OD with Ceramic filtration diluted 10 times
*3.5 Particle size distribution (PSD):*
Concerning a variety of disperse materials, the most important of all physical parameters is the
particle size. The determination of the particle size (granulometric composition) is usually carried out
in a wide range of industries and often this parameter is critically important in the production of a
large number of products. The particle size has a direct influence on Stability in suspension to
determine the polydization of real dispersions and emulsions, we use the well-known concept of
"particle size distribution" (PSD). The results from particle size distribution are shown in tables 6, 7
and 8, charts 5, 6 and 7.
Table 6: The result of experiment from sewage water

| Summary Data | | Percentiles | | Size Percent | | | | |
|---|---|---|---|---|---|---|---|---|
| MI(nm): | 0 | **%Tile** | **Size(nm)** | **Size(nm)** | **%Tile** | | | |
| MN(nm): | 0 | 10.00 | 3050 | 10000 | 100.00 | | | |
| MA(nm): | 0 | 20.00 | 3310 | 20000 | 100.00 | | | |
| CS: | 1.563 | 30.00 | 3500 | 30000 | 100.00 | | **Peaks** | |
| SD: | 783.0 | 40.00 | 3680 | 40000 | 100.00 | **Dia(nm)** | **Vol %** | **Width** |
| PDI: | 0.0780 | 50.00 | 3870 | 50000 | 100.00 | 3870.0000 | 100.00 | 1566.0000 |
| Mz: | 0 | 60.00 | 4070 | 60000 | 100.00 | | | |
| si: | 795.9 | 70.00 | 4320 | 70000 | 100.00 | | | |
| Ski: | 212.0 | 80.00 | 4610 | 80000 | 100.00 | | | |
| Kg: | 0 | 90.00 | 5110 | 90000 | 100.00 | | | |
| | | 95.00 | 5540 | 95000 | 100.00 | | | |



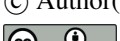




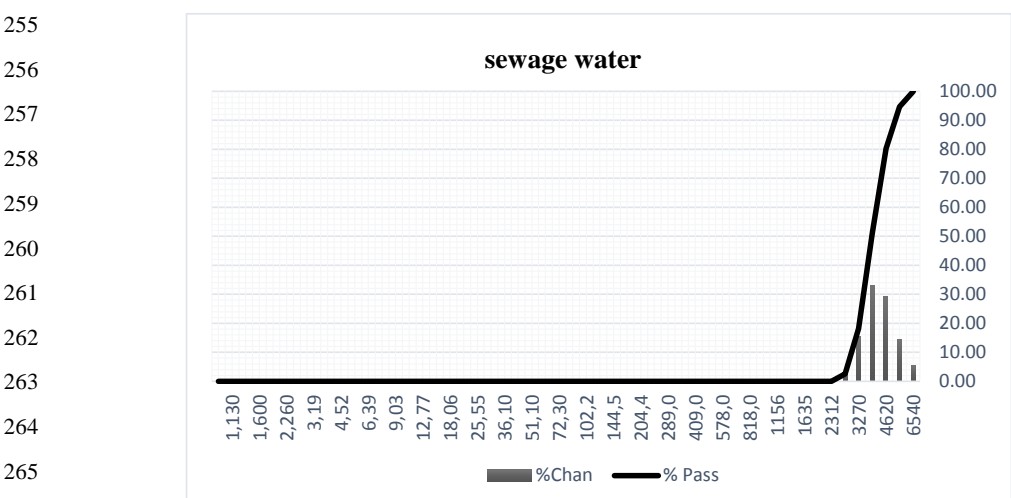

Fig 5: The result of experiment from sewage water

Table 7: The result of experiment from Modified perlite filtration

| Summary Data | | Percentiles | | Size Percent | | | | |
|---|---|---|---|---|---|---|---|---|
| MI(nm): | 0 | **%Tile** | **Size(nm)** | **Size(nm)** | **%Tile** | | | |
| MN(nm): | 92.50 | 10.00 | 186.4 | 10000 | 100.00 | | | |
| MA(nm): | 383.0 | 20.00 | 273.2 | 20000 | 100.00 | | | |
| CS: | 15.65 | 30.00 | 367.0 | 30000 | 100.00 | **Peaks** | | |
| SD: | 617.0 | 40.00 | 456.0 | 40000 | 100.00 | Dia(nm) | Vol % | Width |
| PDI: | 2.6970 | 50.00 | 560.0 | 50000 | 100.00 | 5830.0000 | 9.40 | 920 |
| Mz: | 756.2 | 60.00 | 742.0 | 60000 | 100.00 | 1299.0000 | 24.80 | 505. |
| si: | 0 | 70.00 | 1075 | 70000 | 100.00 | 408.0000 | 62.40 | 172.9 |
| Ski: | 663.1 | 80.00 | 1350 | 80000 | 100.00 | 65.6000 | 3.40 | 23.48 |
| Kg: | 0 | 90.00 | 1845 | 90000 | 100.00 | 5830.0000 | 9.40 | 920 |
| | | 95.00 | 5790 | 95000 | 100.00 | | | |










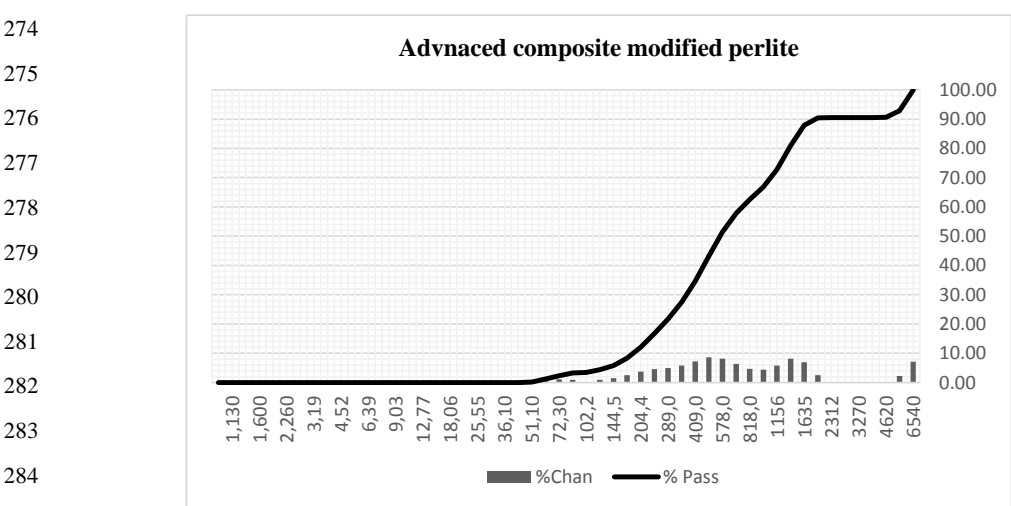

Fig 6: The result of experiment from Modified perlite filtration

Table 8: The result of experiment from Ceramic filtration

| Summary Data | | Percentiles | | Size Percent | | | | |
|---|---|---|---|---|---|---|---|---|
| MI(nm): | 977.0 | **%Tile** | **Size(m)** | **Size(nm)** | **%Tile** | | | |
| MN(nm): | 92.50 | 10.00 | 266.9 | 10000 | 100.00 | | | |
| MA(nm): | 367.0 | 20.00 | 343.0 | 20000 | 100.00 | | | |
| CS: | 16.36 | 30.00 | 378.0 | 30000 | 100.00 | **Peaks** | | |
| SD: | 154.5 | 40.00 | 407.0 | 40000 | 100.00 | Dia(nm) | Vol % | Width |
| PDI: | 2.0880 | 50.00 | 433.0 | 50000 | 100.00 | 4050 | 15.40 | 1720 |
| Mz: | 461.6 | 60.00 | 459.0 | 60000 | 100.00 | 419.0 | 80.40 | 431 |
| si: | 729.2 | 70.00 | 493.0 | 70000 | 100.00 | 69.1 | 4.20 | 30.2 |
| Ski: | 575.7 | 80.00 | 553.0 | 80000 | 100.00 | | | |
| Kg: | 0 | 90.00 | 3720 | 90000 | 100.00 | | | |
| | | 95.00 | 4470 | 95000 | 100.00 | | | |
















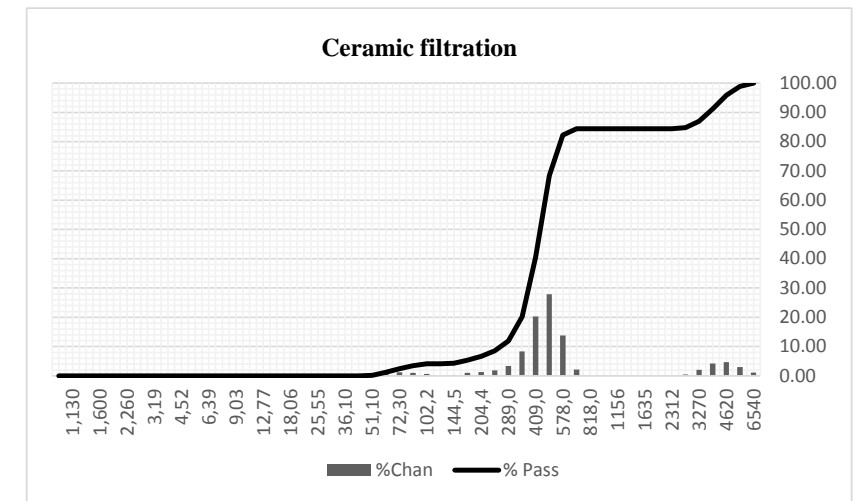

Fig 7: The result of experiment from Ceramic filtration

**4. Economic view of project:**
The view that has emerged is that the comparing the cost of using ceramic filter to remove turbidity
and 3 steps filtration with membrane and modified perlite. And quality index has been created to
compare both cost and performance among each. The results of economic view is shown in table9.

|  | Cost | performance | Quality index(i=2)= $\dfrac{(Performance)^i}{Cost\$}$ |
|---|---|---|---|
| Modified perlite | 115$ | 99% | 0.8522 |
| Ceramic Filtration | 140$ | 94% | 0.6311 |


**5. Discussions:**
The significant decrease on COD which can be easily found it was because of using coagulants in the perlite in
sodium silicate perlite which is able to reduce COD, as it is expected. Furthermore, synthetic zeolite has
considerable role in reducing TOC in wastewater in general, therefore technically TOC has decreased due to the
fact. The perlite itself is able to decrease turbidity of water, the proposed system is consist of 4 kinds of different
perlite in different sizes which lead to obliterate turbidity.



In addition of removing turbidity of water, optical density which is able to measure a tiniest elements in the
water, demonstrates the hereinbefore fact. The results from particle size distribution attest the function of
advanced composite modified perlite.

## 6. Conclusions:

Using the device with 2 steps siphon, due to having sedimentation part in bottom part of next part
and having two times upward direction in filtration works better than monolayer filter with only
perlite. Concerning optical density, the advanced composite modified perlite which ought to be used
through 2 step filtration device is able to notably diminish the optical density from 0.36 to 0.009
whereas the ceramic filtration decreases from 0.36 to 0.086 in wave length range in 254. In other
word, the advanced composite modified perlite plummets optical density up to 97.5% which its
counterpart drops the optical density merely 76.1%.
Modified perlite filtration is able to obtain TN less than 1 mg/L from 3.81 mg/L to 0.87 mg/L,
decrease COD up to 99% from 274 to 0.42 mgO/L and decrease particle size up to 98% from 3870 nm
to 56 nm.
Concerning PSD, from the beginning the maximum diameter of particle size in the sewage water
was 3870 nm with 1566 width. Subsequent to filtering sewage water with advanced composite
modified, the maximum diameter of particle size which has the majority of 62.4% was 408 nm and the
width was 172.9, the data for its counterpart measured as maximum diameter of particle size which
has the majority of 80.4% was 419.0 nm with the 431 width.
The required pressure for this filtration is 1 atmosphere which was required for the ceramic filtration
as well.
The advanced composite modified perlite give rise to adjust the pH of sewage water.
Total cost of filtration unit containing whole part of filtration device and modified perlite materials as
well as activated carbon inside is less by 12% compared to ceramic filter.

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
