# Peer review of "Application of advanced composite modified perlite for degradation of particle size and turbidity in treatment of sewage water"

_Drinking Water Engineering and Science, 2017_

## Referee Comment (RC1) · Anonymous Referee #1 · 10 Jan 2018

Application of advanced composite modified perlite for degradation of particle and turbidity in treatment of sewage water

It is a study where sewage water have been filtered through perlite whereby the turbidity is reduced. The method is already well-known and used e.g. for cleaning water in swimming pools. No explanation of the experimental setup is presented, so I cannot redo the experiment, e.g. no data on the amount of perlite used, the filtration column, filtration time. Is the filter backwashed and how often is the material removed and new material used.

There are many linguistic errors. Titel: Application of ... perlite ... used for degradation

of particle size ..." No particle size is not degradated, some particles are removed.

Line 29-30 page1: "To meet the need of people and to provide access to clean water and, in the twentieth century is difficult and complicated. Global growth in the public water supply, improve water quality, climate change and are growing rapidly" do not make any sense. The paper is difficult to read.

Many unnecessary information is presented e.g. details on the different type of materials e.g. page 4 line 107-108: For the first time liquid glass in 1818 was received by the german chemist Jan Nepomuk von Fuchs. Just write where the material come from and the chemical position and physico-chemical properties of the materials.

Split Experiments and results section in two (1. Experiments and 2. Results.)

Fig 2 - 4 - Change so wave length is on the X-axis and Absorbance or OD on the Y-axis

Fig 5 and 6: What is on the two axes?

In my opinion the paper is not good enough for publication.

---

## Author Comment (AC1) · 13 Jan 2018

1. according to a well-known method for using perlite in a swimming pool, the usage is merely limited to the simple perlite. in my article, I modified the perlite by using sodium silicate through a certain method which will be a patent and creating synthetic zeolite which gets combined with the perlite. the materials have been made new in this area of work. that's not fair to compare it with pelite using in swimming pool. 2. in this experiment, "degrade" means to reduce the particle which can be detected from particle size distribution machine. 3. focus is on how the materials function, in fact, the process of making modified perlite and quantity of usage perlite is recorder which I will

write it down in a revised version. 4. regarding Fig2, I changed it, the device showed a result wave length is on X-axis and OD is on the Y-axis. 5. regarding Fig5 and 6, X-axis is size (nanometer, and Y-axis is Channel%, Pass%. best regard

---

## Referee Comment (RC2) · Anonymous Referee #2 · 12 Feb 2018

The aim of this paper was to study a sewage-water treatment performance of a filtering device, filled with modified perlite, on COD, turbidity, etc., removal efficiency. The content of this paper could contribute to the field of wastewater treatment. However, I have some concerns about the English used and the limited information given in results and discussion sections. I am recommending acceptance of this paper for publication after the following comments are addressed by the authors.

Comments: 1. Abstract: "Water treatment efficiency" or "Wastewater treatment efficiency"? 2. Abstract: "concentration and size of particles (from 3870 nm to 56 nm)". Which is the concentration of the particles? 3. Abstract: "Filtering device was created

with 2 steps syphon, . . . filter with mere perlite.". Please rewrite the sentence. 4. Introduction: "To meet the needs of people and to provide access to clean water and, in the twentieth century is difficult and complicated." Please correct the sentence. 5. Introduction: "Advantages of perlite as a filter aid include reducing the cost with the help of perlite filter which implies the aid of perlite filter is 20 to 50% more than other filters, which is very significant in terms of cost reduction." This sentence is confusing. 6. Introduction: "Application of perlite in filtration process has not been used for over a decade, . . ... in scientific area of filtration." This sentence is quite to long and confusing. 7. The goal of this study is not well presented. 8. Throughout the manuscript - please correct English typo and grammar mistakes. 9. Table 1: please correct the formulas. 10. 2.2 SC Perlite: Please give a reference to support the information provided in this section. 11. 2.4 Perite with modified Fuller'curve: "I used modified Fuller Thompson Curve.." Correct the typo mistakes and avoid writing in first person. 12. 3.1 Turbid meter (NTU): Describe the method used for turbidity. 13. 3.2 TOC and 3.3 Nitrate: It seems that authors described in general the meaning of Nitrates and TOC presence, without explaining their results. 14. Economic view of project: The information given in this section is limited.

---

## Author Comment (AC2) · 13 Feb 2018

Dear Referee thanks for your kind comments. I will take your advice into my article, and I will make modifications to the grammatical mistakes which give rise to the misunderstanding of the sentences. moreover, I take your advice into account in order to boost my article's quality. best regard